# Competition and resource depletion shape the thermal response of population fitness in *Aedes aegypti*

Paul J. Huxley [1✉], Kris A. Murray[1,2], Samraat Pawar[3] & Lauren J. Cator [3]

Mathematical models that incorporate the temperature dependence of lab-measured life history traits are increasingly being used to predict how climatic warming will affect ectotherms, including disease vectors and other arthropods. These temperature-trait relationships are typically measured under laboratory conditions that ignore how conspecific competition in depleting resource environments—a commonly occurring scenario in nature—regulates natural populations. Here, we used laboratory experiments on the mosquito *Aedes aegypti*, combined with a stage-structured population model, to investigate this issue. We find that intensified larval competition in ecologically-realistic depleting resource environments can significantly diminish the vector's maximal population-level fitness across the entire temperature range, cause a ~6 °C decrease in the optimal temperature for fitness, and contract its thermal niche width by ~10 °C. Our results provide evidence for the importance of considering intra-specific competition under depleting resources when predicting how arthropod populations will respond to climatic warming.

[1] MRC Centre for Global Infectious Disease Analysis, School of Public Health, Imperial College London, London, UK. [2] MRC Unit The Gambia at London School of Hygiene & Tropical Medicine, Banjul, The Gambia. [3] Department of Life Sciences, Imperial College London, Ascot, UK. ✉email: p.huxley@imperial.ac.uk

Global environmental change is predicted to affect the spatiotemporal distributions of arthropods, including disease vectors and the diseases they transmit[1,2]. For example, a recent study suggests that climatic warming may increase the thermal suitability for Zika virus transmission, leading to 1.3 billion more people being at risk of exposure by 2050[3]. Other studies have predicted that warming will increase the global invasion potential of *Aedes aegypti*, a principal vector of dengue, yellow fever and chikungunya[4]. Such predictions typically arise from mathematical models that incorporate thermal performance curves (TPCs) for vector life history traits, such as juvenile development and mortality, which together define the TPC of maximal population growth rate ($r_m$, a measure of population fitness)[5].

Typically, such trait-level TPC data come from larval populations reared under optimal food conditions in the laboratory (e.g.,[6]). However, recent studies suggest that many predictions of how vector populations will respond to climatic warming are likely to be biased. For example, when food is supplied at a constant rate, low resource availability in the larval stage can have a significant negative effect on temperature-trait relationships[7–9]. In particular, our recent work[9] has shown that low-resource supply, through its adverse impact on juvenile traits, can significantly depress population fitness and decrease its predicted peak temperature. Despite such advances, resources in natural habitats are not constant and, in many, or arguably even most cases, deplete over time[10–12]. For example, *Ae. aegypti* is expected to be strongly regulated by conspecific competition between larvae[13,14], because this stage of the species' lifecycle is confined to small isolated water bodies that are susceptible to infrequent resource inputs and, therefore, resource depletion[15–18].

In such small, isolated aquatic habitats, resource levels will deplete if consumption rates exceed replacement rates. Abiotic factors such as rainfall may also abruptly dilute resources in the habitat, exacerbating the biotic resource depletion rate. The extent of resource depletion ultimately influences the strength of larval competition in mosquitoes, acting as a regulatory mechanism on the population. In particular, as per-capita energy requirements increase with warming, resource depletion from direct consumption, and therefore the strength of competition should also increase. These combined effects are bound to compromise the development and survival of individuals as the deficit between resource uptake and energy use increases. These trait-level effects are then expected to propagate through the stage-structured population dynamics to affect the shape of the $r_m$ TPC[19,20]. This is because $r_m$ is essentially proportional to the difference between biomass gained through consumption and that lost to respiration and mortality[5]. Moreover, intensified competition should decrease $r_m$ across temperatures, albeit to different degrees.

Furthermore, if the rate of biomass loss increases faster than any increase in biomass gain with temperature, the thermal optimum of ($r_m$ $T_{opt}$) may also shift downwards[21,22]. For the same reason, the range of temperatures over which $r_m$ is positive (the thermal niche width) may become narrower. As a result, the combined effects of climatic warming and decreased resource availability could contribute to the contraction of species range boundaries. This effect could simultaneously decrease the burden of vector-borne diseases and agricultural pests but increase the extinction risk of vulnerable species[23,24]. Conversely, concurrent increases in temperature and resource availability with climatic warming could have the opposite effect by optimising $r_m$, and thus, promoting the invasion and establishment of tropical taxa into temperate habitats[25]. This effect could further increase the huge socioeconomic cost of invasions by disease vectors, such as *Aedes* mosquitoes[26].

Studies across a broad range of taxa are needed to make generalisable predictions on the ecological impacts of environmental change on ectotherm populations, including disease vectors[27,28]. So far, however, the effects of competition in depleting resource environments on the temperature dependence of ectotherm fitness have mainly focused on single-celled prokaryotes[29–31]. To address this important deficit, we investigated the effects of competition on the $r_m$ TPC by exposing *Ae. aegypti* larvae to an ecologically realistic range of temperatures and depleting resource levels. We show that competition in resource depletion scenarios will significantly change the shape of the thermal response of mosquito population fitness—key for predicting how disease vectors and other arthropods will respond to environmental change. Our findings allow us to infer that there are thresholds of resource availability, below which intensifying competition causes a dramatic change in this temperature dependence of fitness.

## Results

We investigated how *Ae. aegypti* population fitness traits respond to temperature and resource depletion using a factorial experimental design comprised of five temperatures and four resource levels. We used standard linear model (LM) fitting to analyse normally distributed trait responses (adult lifespan and body size). For trait data that were not normally distributed (juvenile development time), we used generalised linear model (GLM) fitting to analyse these responses. For juvenile mortality, we fitted an exponential function to survival data using the 'flexsurv' R package[32].

All trait responses varied significantly with temperature and resource level, with a significant interaction between the two environmental variables (Fig. 1, Tables 1, 2).

Larval competition at our lowest resource level (0.183 mg ml$^{-1}$) increased the negative effect of increased temperature on juvenile mortality rate (Fig. 1a, Table 2). As temperatures increased from 22 to 34 °C, non-overlapping 95% credible intervals indicate that juvenile mortality rate was significantly higher at low-resource levels than at intermediate-resource levels (0.367 mg ml$^{-1}$). At 0.183 mg ml$^{-1}$, it increased by ~200% from 0.05 at 22 °C to 0.14 individual$^{-1}$ day$^{-1}$ at 34 °C. In contrast, at 0.367 mg ml$^{-1}$, the juvenile mortality rate increased by 25% (from 0.04 to 0.05 individual$^{-1}$ day$^{-1}$) across this temperature range.

The interaction between temperature and resource level caused significant variation in development time across treatments (ANOVA; $F_{9, 2.24} = 13.44$, $P < 0.001$, Table 1). Development time decreased with temperature at all resource levels, but the decrease with temperature was greater at the low resource level than at higher resource levels due to resource depletion (Fig. 1b). At 0.183 mg ml$^{-1}$, development time decreased from 18.30 days at 22 °C to 8.26 days at 34 °C. Development time at the higher resource levels decreased from ~13.50 days at 22 °C to ~7.50 days at 34 °C (Table 2).

Competition at low resource levels (0.183 mg ml$^{-1}$) resulted in significant variation in size at maturity (mass, mg) between resource levels (ANOVA; $F_{9, 0.92} = 24.26$, $P < 0.001$, Table 1). Adult size decreased both at warmer temperatures and at low-resource levels, though the decrease with temperature was greater at higher resource levels than at the low resource level. At low-resource levels, size decreased by 0.13 mg as temperatures increased from 22 to 34 °C, while at the highest resource level (0.733 mg ml$^{-1}$), size decreased by 0.26 mg (Fig. 1c, Table 2).

The interaction between temperature and resource level caused significant variation in adult lifespan across treatments (ANOVA; $F_{9, 699.60} = 7.96$, $P < 0.001$, Table 1). The adult size was largest at the highest larval resource level (0.733 mg ml$^{-1}$) at 22 °C and 26 °C, which caused lifespan to be greatest at these temperatures (11.24 and 11.65 days, respectively). Lifespan at 0.733 mg ml$^{-1}$ then decreased to 4.68 days at 34 °C. In contrast, at low resource

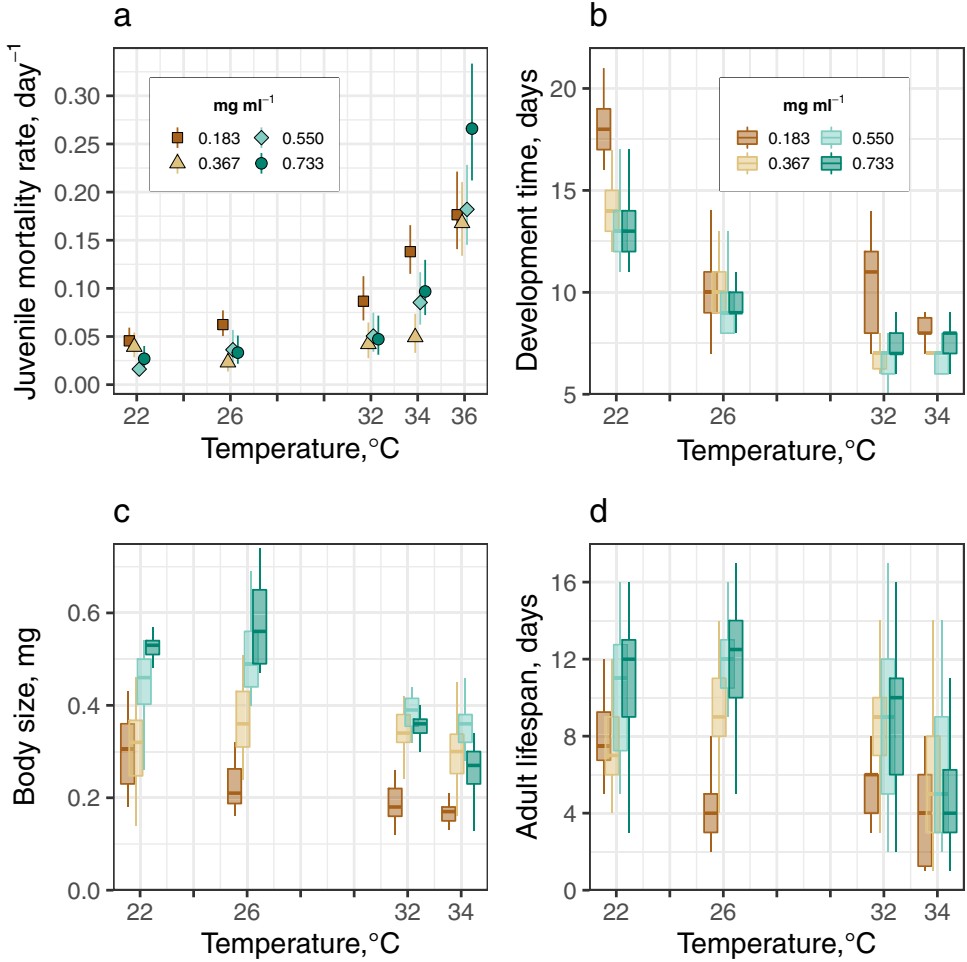

**Fig. 1 The effect of larval competition on fitness traits in *Ae. aegypti*. a** Competition at low resource levels (0.183 mg ml$^{-1}$) increased the negative effect of increased temperature on juvenile mortality. Symbols with 95% confidence intervals denote the predicted mortality rates for each treatment derived from fitting an exponential function to our survival data using the 'flexsurv'[32] R package. **b** Development time decreased with temperature at all resource levels but, at most temperatures, it was significantly extended by competition at 0.183 mg ml$^{-1}$. **c** As temperatures increased from 22 °C, competition at 0.183 mg ml$^{-1}$ significantly reduced size at emergence. **d** As temperatures increased from 22 to 32 °C, competition at 0.183 mg ml$^{-1}$ significantly reduced adult lifespan. The resulting ANOVAs of the regressions for each trait are presented in Table 1. Boxplot horizontal lines represent medians. Lower and upper hinges are the 25th and 75th percentiles. Upper whiskers extend from the hinge to the largest value no further than 1.5 × inter-quartile range (IQR) from the hinge. The lower whisker extends from the hinge to the smallest value at most 1.5 × IQR of the hinge. The number of female mosquitoes in each treatment is shown in Table 2. Source data are in Supplementary Data 1.

levels, decreased size at maturity caused lifespan to decrease from 8.00 days at 22 °C to 3.76 days at 34 °C mg (Fig. 1d, Table 2).

At all resource levels, predicted daily fecundity rate increased with temperature (Table 2), though the increase was greater at the lowest resource level than at higher resource levels. At the lowest resource level, fecundity increased with temperature from 1.77 eggs individual$^{-1}$ day$^{-1}$ at 22 °C to 6.87 eggs individual$^{-1}$ day$^{-1}$ at 34 °C. At the higher resource levels, fecundity increased from ~2 eggs at 22 °C to ~5 eggs individual$^{-1}$ day$^{-1}$ at 34 °C.

**Population fitness**. At all resource levels, $r_m$ responded unimodally to temperature. However, intensified larval competition at low resource levels (0.183 mg ml$^{-1}$) significantly depressed $r_m$ across the entire temperature range (Fig. 2a) and caused it to peak at a significantly lower temperature than at intermediate resource levels (0.367 mg ml$^{-1}$; Fig. 2b, Table 3). Competition at 0.183 mg ml$^{-1}$ also significantly narrowed the thermal niche width for $r_m$ compared to higher resource levels (Fig. 2a, Table 3).

At 0.183 mg ml$^{-1}$, $r_m$ was negative until temperatures increased to 23.3 °C (Fig. 2, Table 3). At this resource level, $r_m$

reached a peak of 0.05 at its $T_{opt}$ (26.6 °C); it then declined to negative growth at 30.1 °C. The breadth of $r_m$'s thermal niche width at the lowest resource level was 6.8 °C. In contrast, at the intermediate food level (0.367 mg ml$^{-1}$), $r_m$ became positive as temperatures increased to 18.8 °C; it was maximal at 33.0 °C (0.24, Fig. 2, Table 3). At 0.367 mg ml$^{-1}$, $r_m$ declined to negative growth at 35.4 °C. The thermal niche width for $r_m$ at this resource level was 16.6 °C. Overlapping CIs indicate that the predicted differences between the intermediate resource level and the higher resource levels (0.550 and 0.733 mg ml$^{-1}$) in $r_m$ at $T_{opt}$, $T_{opt}$, and the thermal niche width were non-significant (Fig. 2, Table 3).

**Sensitivity analyses**

*Elasticities*. Juvenile traits (development time and survival) contributed more substantially to $r_m$ than adult traits (Fig. 3). For example, at the lowest resource level (0.183 mg ml$^{-1}$) at 26 °C, a 0.5 proportional increase in juvenile traits would increase the rate of increase from 0.046 to 0.063 (Fig. 3d). By contrast, for the same treatment, increases of equal proportions in adult survival and fecundity would increase $r_m$ from 0.046 to 0.050 (Fig. 3e) and

from 0.046 to 0.048 (Fig. 3f), respectively. This highlights how the temperature-dependence of $r_m$ stems mainly from how competition impacts juvenile survival and development. Juvenile survival determines the number of reproducing individuals, whereas, development rate governs the timing of reproduction. The carry over effect of reduced size at maturity on $r_m$ is relatively weak, because fecundity and adult survival have comparatively small effects on $r_m$.

*Fecundity estimates.* Figure 4 shows that the $r_m$ TPCs were insensitive to uncertainty in our fecundity estimates. Comparison with the central estimates shows that, for all resource levels, using the upper and lower 95% exponents (Supplementary Eq. 1; Supplementary Fig. 1b) for the scaling between lifetime fecundity and size does not qualitatively change the predicted $r_m$ TPCs, or the matrix projection $r_m$ estimates that were used to fit the $r_m$ TPCs. Predicted $r_m$ $T_{opt}$ was also insensitive to uncertainty in our fecundity estimates. Also, using the upper and lower 95% exponents (Supplementary Eq. 1; Supplementary Fig. 1b) for the scaling between lifetime fecundity and size does not qualitatively change predicted maximal $r_m$ or $r_m$ $T_{opt}$.

## Discussion

Global climate change is expected to have far-reaching impacts on the distributions and abundances of ectotherms, prompting calls for greater understanding of how density dependent and density independent factors interact to regulate their population fitness. So far, studies on the effects of interactions between resource concentration and temperature on fitness in ectotherms have mainly focused on single-celled organisms[29–31]. Studies on eukaryotic ectotherms have shown that their population fitness is inversely related to intensified competition between larvae for depleting resources[13,33]. However, barring a few notable exceptions[34], such studies have not generally included temperature. Here, we have shown that larval competition can significantly change the shape of the $r_m$ thermal response in *Ae. aegypti*. We also show that there are resource availability thresholds, below which competition intensifies, causing a dramatic change in the temperature dependence of fitness. Together, our findings indicate that competition in depleting resource environments is an important regulatory mechanism that needs to be considered when predicting how organisms with complex life cycles will respond to anticipated shifts in environmental temperature with global change.

At the lowest resource level (0.183 mg ml$^{-1}$), competition had a consistent negative effect on the thermal responses of

**Table 1 Type II Analysis of Variance results from regression models fitted to the responses of life history traits to temperature and resource level (RL).**

| Trait | Predictor | $\chi^2$ | df | F value | P value |
|---|---|---|---|---|---|
| Development time (GLM) $R^2 = 0.83$ | **Temperature** | **50.28** | **3** | **903.40** | **<0.001***** |
| | **RL** | **3.97** | **3** | **71.28** | **<0.001***** |
| | **Temperature × RL** | **2.24** | **9** | **13.44** | **<0.001***** |
| | Replicate | 0.07 | 2 | 1.77 | 0.17 |
| | Residuals | 12.63 | 681 | | |
| Adult lifespan (LM) $R^2 = 0.41$ | **Temperature** | **1594.80** | **3** | **54.44** | **<0.001***** |
| | **RL** | **1908.60** | **3** | **65.15** | **<0.001***** |
| | **Temperature × RL** | **699.60** | **9** | **7.96** | **<0.001***** |
| | Replicate | 15.70 | 2 | **0.81** | 0.45 |
| | Residuals | 6533.10 | 669 | | |
| Body size (LM) $R^2 = 0.73$ | **Temperature** | **1.32** | **3** | **104.92** | **<0.001***** |
| | **RL** | **2.58** | **3** | **204.38** | **<0.001***** |
| | **Temperature × RL** | **0.92** | **9** | **24.26** | **<0.001***** |
| | Replicate | 0.02 | 2 | 2.31 | 0.10 |
| | Residuals | 1.81 | 431 | | |

Significant effects are shown in boldface type.
*P value < 0.05; **P value < 0.01; ***P value < 0.001.

**Table 2 Comparison of the effect of larval competition on the temperature-dependence of population fitness ($r_m$) and its component traits.**

| Trait | Temperature (°C) | Resource level (mg ml$^{-1}$) Mean ± s.e.m. | | | |
|---|---|---|---|---|---|
| | | 0.183 | 0.367 | 0.550 | 0.733 |
| Development time (days) | 22 | 18.30 ± 0.56 ($n = 20$) | 14.41 ± 0.34 ($n = 34$) | 13.41 ± 0.25 ($n = 54$) | 13.33 ± 0.25 ($n = 51$) |
| | 26 | 10.45 ± 0.18 ($n = 65$) | 10.35 ± 0.20 ($n = 51$) | 9.32 ± 0.19 ($n = 44$) | 9.19 ± 0.17 ($n = 53$) |
| | 32 | 10.11 ± 0.32 ($n = 19$) | 6.98 ± 0.13 ($n = 54$) | 6.78 ± 0.13 ($n = 50$) | 7.19 ± 0.13 ($n = 53$) |
| | 34 | 8.26 ± 0.19 ($n = 34$) | 7.04 ± 0.14 ($n = 51$) | 6.67 ± 0.15 ($n = 36$) | 7.87 ± 0.20 ($n = 30$) |
| Juvenile mortality rate (individual$^{-1}$ day$^{-1}$) | 22 | 0.05 ± 0.01 ($n = 75$) | 0.04 ± 0.01 ($n = 71$) | 0.02 ± 0.00 ($n = 75$) | 0.03 ± 0.01 ($n = 75$) |
| | 26 | 0.06 ± 0.01 ($n = 150$) | 0.02 ± 0.01 ($n = 65$) | 0.04 ± 0.01 ($n = 63$) | 0.03 ± 0.01 ($n = 74$) |
| | 32 | 0.09 ± 0.01 ($n = 75$) | 0.04 ± 0.01 ($n = 75$) | 0.05 ± 0.01 ($n = 75$) | 0.05 ± 0.01 ($n = 75$) |
| | 34 | 0.14 ± 0.01 ($n = 150$) | 0.05 ± 0.01 ($n = 75$) | 0.09 ± 0.01 ($n = 75$) | 0.10 ± 0.01 ($n = 75$) |
| | 36 | 0.18 ± 0.02 ($n = 75$) | 0.17 ± 0.02 ($n = 75$) | 0.18 ± 0.02 ($n = 75$) | 0.27 ± 0.03 ($n = 75$) |
| Adult lifespan (days) | 22 | 8.00 ± 0.70 ($n = 20$) | 7.50 ± 0.54 ($n = 34$) | 10.04 ± 0.43 ($n = 54$) | 11.24 ± 0.45 ($n = 49$) |
| | 26 | 4.54 ± 0.39 ($n = 65$) | 9.39 ± 0.45 ($n = 49$) | 11.51 ± 0.48 ($n = 43$) | 11.65 ± 0.43 ($n = 52$) |
| | 32 | 5.21 ± 0.72 ($n = 19$) | 8.53 ± 0.43 ($n = 53$) | 8.66 ± 0.44 ($n = 50$) | 9.29 ± 0.43 ($n = 52$) |
| | 34 | 3.76 ± 0.54 ($n = 34$) | 5.62 ± 0.44 ($n = 50$) | 6.09 ± 0.53 ($n = 35$) | 4.68 ± 0.59 ($n = 28$) |
| Body size (dry mass (mg)) | 22 | 0.30 ± 0.01 ($n = 20$) | 0.31 ± 0.01 ($n = 32$) | 0.44 ± 0.01 ($n = 52$) | 0.52 ± 0.01 ($n = 46$) |
| | 26 | 0.23 ± 0.01 ($n = 20$) | 0.36 ± 0.01 ($n = 31$) | 0.50 ± 0.01 ($n = 29$) | 0.58 ± 0.01 ($n = 21$) |
| | 32 | 0.19 ± 0.01 ($n = 19$) | 0.34 ± 0.01 ($n = 29$) | 0.38 ± 0.01 ($n = 31$) | 0.36 ± 0.01 ($n = 30$) |
| | 34 | 0.17 ± 0.02 ($n = 9$) | 0.30 ± 0.01 ($n = 30$) | 0.36 ± 0.01 ($n = 27$) | 0.26 ± 0.01 ($n = 23$) |
| Daily fecundity rate (eggs individual$^{-1}$ day$^{-1}$) | 22 | 1.77 ± 0.06 ($n = 20$) | 1.95 ± 0.05 ($n = 32$) | 1.99 ± 0.06 ($n = 52$) | 2.59 ± 0.54 ($n = 46$) |
| | 26 | 1.52 ± 0.07 ($n = 20$) | 1.96 ± 0.08 ($n = 31$) | 2.54 ± 0.37 ($n = 29$) | 3.00 ± 0.30 ($n = 21$) |
| | 32 | 3.79 ± 0.26 ($n = 19$) | 3.04 ± 0.33 ($n = 29$) | 3.55 ± 0.51 ($n = 31$) | 2.64 ± 0.22 ($n = 30$) |
| | 34 | 6.87 ± 1.87 ($n = 9$) | 5.07 ± 0.82 ($n = 30$) | 4.91 ± 0.52 ($n = 27$) | 5.76 ± 0.89 ($n = 23$) |
| Population-level fitness ($r_m$) | 22 | −0.05 ± 0.03 | 0.06 ± 0.02 | 0.10 ± 0.01 | 0.08 ± 0.02 |
| | 26 | 0.05 ± 0.02 | 0.12 ± 0.02 | 0.14 ± 0.02 | 0.16 ± 0.02 |
| | 32 | −0.04 ± 0.04 | 0.24 ± 0.02 | 0.23 ± 0.03 | 0.20 ± 0.03 |
| | 34 | −0.21 ± 0.05 | 0.22 ± 0.03 | 0.15 ± 0.04 | 0.02 ± 0.05 |
| | 36 | −2.10 ± 0.15 | −2.11 ± 0.21 | −2.19 ± 0.22 | −2.68 ± 0.20 |

The means with standard errors for juvenile mortality rate were estimated by fitting an exponential function to survival data for each treatment using the 'flexsurv' R package. The means with standard errors for development time, lifespan and size were estimated by using the statistical models in Table 1 (replicate dropped). For fecundity, the standard errors were estimated using the 'Rmisc' package in R. For $r_m$, 95% CIs were approximated using the delta method[75]. The number of individual mosquitoes is shown in parentheses ($n =$) after the means for each treatment. For $r_m$ TPC fitting, non-positive matrix projection $r_m$ values at 36 °C were adjusted to −0.30. For plotting (Fig. 2a), non-positive $r_m$ values were cut off at −0.10.

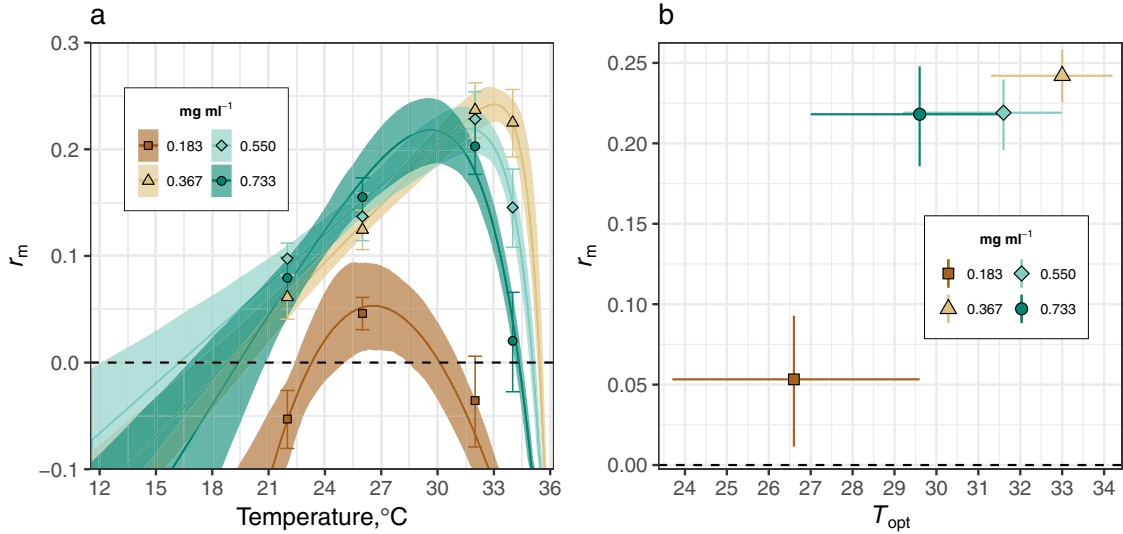

**Fig. 2 The effect of larval competition on the thermal response of population-level *Ae. aegypti* fitness ($r_m$) with bootstrapped 95% prediction bounds.**
**a** Competition at 0.183 mg ml$^{-1}$ significantly depressed $r_m$ across the entire temperature range and narrowed its thermal niche width compared to the higher resource levels (non-overlapping 95% prediction bounds, Table 3). Symbols denote matrix projection estimates with 95% confidence intervals (Table 2). **b** Competition at 0.183 mg ml$^{-1}$ significantly (non-overlapping 95% confidence intervals) lowered maximal $r_m$ and caused it to peak ($r_m$ $T_{opt}$) at a significantly lower temperature than at the intermediate resource level (0.367 mg ml$^{-1}$). Predicted $r_m$ $T_{opt}$ at 0.183 mg ml$^{-1}$ indicates that competition could decrease $r_m$ $T_{opt}$ by 6.4 °C, when compared to the intermediate resource level (0.367 mg ml$^{-1}$, Table 3). The Kamykowski model[82] (Eq. 2) predictions are in Supplementary Data 2.

---

**Table 3 Parameter estimates of the thermal performance curves of population fitness ($r_m$) by resource level.**

| Resource level (mg ml$^{-1}$) | $r_m$ at $T_{opt}$ (± 95% CI) | $T_{opt}$ (°C) (95% CI) | $T_{min}$ (°C) (95% CI) | $T_{max}$ (°C) (95% CI) | Thermal niche width (°C) (95% CI) |
|---|---|---|---|---|---|
| 0.183 | 0.05 ± 0.04 | 26.6 (23.7–29.6) | 23.3 (22.4–24.9) | 30.1 (28.6–31.2) | 6.8 (3.7–8.8) |
| 0.367 | 0.24 ± 0.02 | 33.0 (31.3–34.2) | 18.8 (17.1–20.2) | 35.4 (35.4–35.7) | 16.6 (15.2–18.6) |
| 0.550 | 0.22 ± 0.02 | 31.6 (29.2–33) | 16.2 (12.4–18.6) | 35.1 (35.0–35.3) | 18.8 (16.4–22.9) |
| 0.733 | 0.22 ± 0.03 | 29.6 (27.0–31.5) | 19.4 (16.6–21.0) | 34.3 (34.2–34.6) | 14.9 (13.2–18) |

Non-overlapping 95% confidence intervals (CIs) indicate that larval competition at the lowest resource level (0.183 mg ml$^{-1}$) significantly depressed maximal growth ($r_m$ at $T_{opt}$) compared to the higher resource levels. Competition at 0.183 mg ml$^{-1}$ caused a significant decrease in $r_m$ $T_{opt}$ compared to $r_m$ $T_{opt}$ at 0.367 mg ml$^{-1}$; it also caused a significantly narrower thermal niche width (the thermal maximum for fitness, $T_{max}$, minus the thermal minimum, $T_{min}$) compared to the higher resource levels.

---

underlying fitness traits (Fig. 1), which caused a marked divergence between the $r_m$ TPCs (Fig. 2). Competition at the lowest resource level significantly depressed $r_m$ across the entire temperature range, caused a significant decrease (~6 °C) in $r_m$ $T_{opt}$ compared to the intermediate resource level (0.367 mg ml$^{-1}$), and led to a ~180% contraction of the $r_m$ thermal niche width compared to the higher resource levels (Fig. 2, Table 3).

The elasticity analysis shows that the key mechanism underlying the divergent temperature dependence of $r_m$ across resource levels is increased juvenile development time and mortality at low resource levels (Fig. 3). The negative effect of competition at low-resource levels on these traits delayed the onset of reproduction and population-level reproductive output, respectively. This finding--that juvenile traits contribute more to $r_m$ than adult traits--is consistent with general studies of fitness in organisms with complex lifecycles[35–38], including mosquitoes[9,39].

Furthermore, individual fecundity rate and adult lifespan had negligible effects on $r_m$ compared to juvenile traits, suggesting that the carry over effect of reduced size at maturity on $r_m$ is relatively weak (Fig. 3). For example, at low-resource levels, lifetime fecundity was greater at 22 °C than at 26 °C because body size and adult lifespan were greater at 22 °C. Despite this difference, $r_m$ at 26 °C was predicted to be ~200% greater than at 22 °C (Figs. 1 and 2, Table 2). This result derives from how juvenile development time almost halved as temperatures increased from

22 to 26 °C (Table 2). Although juvenile mortality rates for these treatments were similar (0.05 at 22 °C versus 0.06 at 26 °C, Table 2), faster development at 26 °C meant that greater numbers of individuals could contribute to population growth through reproductive output. This finding is consistent with recent studies that have used constant resource supply rates[9] in suggesting that most projections of how warming will affect disease transmission, through its effects on vector abundance, are likely to be biased because they are likely to underestimate the effect of temperature on juvenile traits and overestimate its effect on adult traits.

While $r_m$ was most sensitive to the interactive effects of temperature and intensified competition on juvenile traits, the carry over effects of this interaction may influence traits that are more directly involved in disease transmission. For example, increased temperature and intensified competition in the larval stages is likely to cause substantial decreases in vectorial capacity by yielding smaller adults that are less likely than larger individuals to outlive to the pathogen's extrinsic incubation period ([40], but see ref. [41]). Future studies could examine how transmission risk changes in response to how temperature interacts with resource depletion to influence other components of vector-borne disease dynamics[42].

Studies on how constant high-resource supply rates affect the temperature dependence of $r_m$ in *Ae. aegypti* similarly report that the optimal temperature for growth in this species is ~32 °C[9,43].

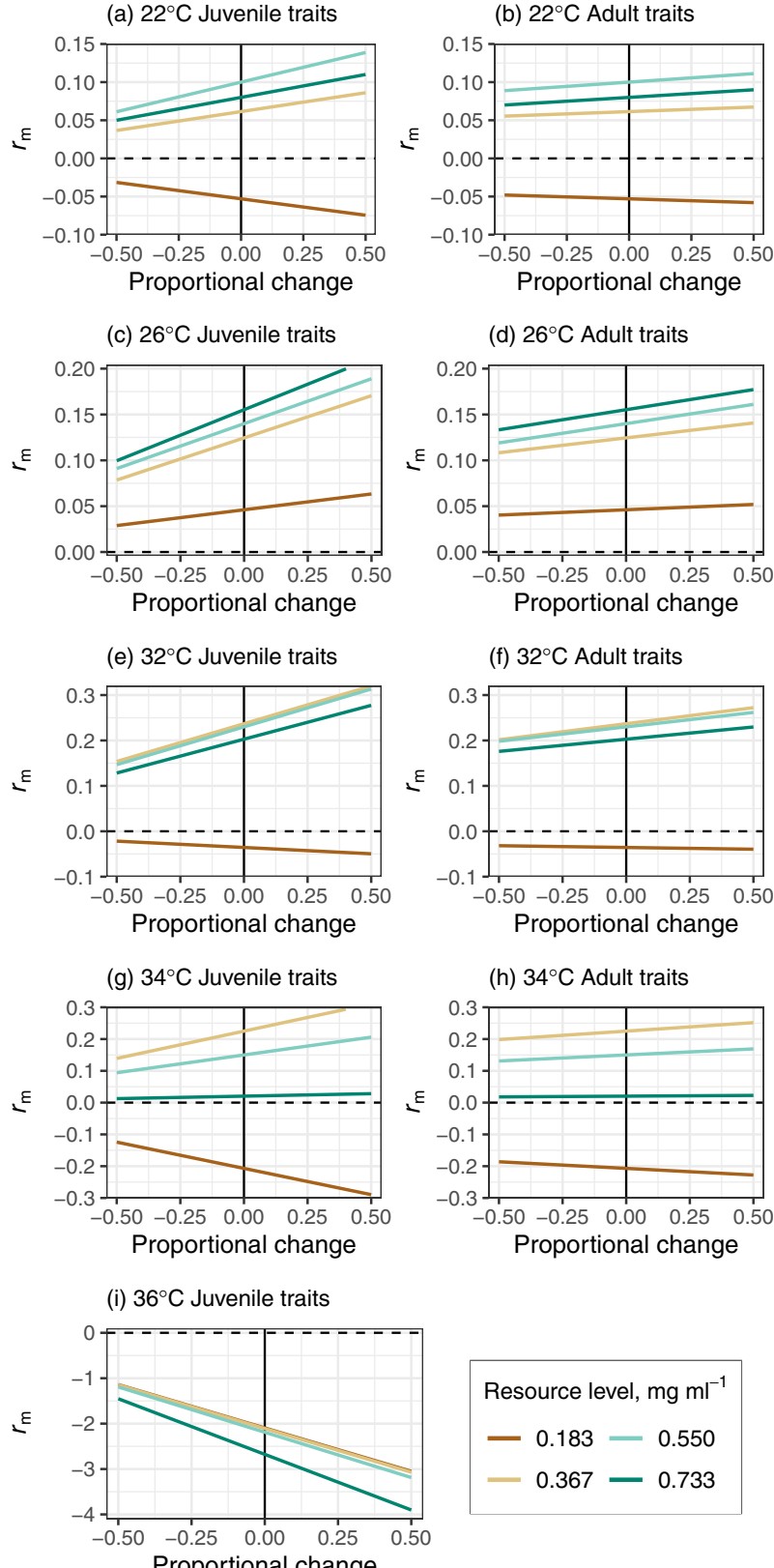

**Fig. 3 Sensitivity of *Ae. aegypti* population fitness ($r_m$) to proportional changes in juvenile and adult traits by temperature across resource levels.**
Juvenile survival and development contributed more substantially to $r_m$, as relatively small changes in the summed matrix elements for these traits would result in relatively large changes in $r_m$. Sensitivity of $r_m$ to adult traits (survival, fecundity) was much weaker compared to sensitivity to juvenile traits. Juvenile traits are shown in (**a**), (**c**), (**e**), (**g**), and (**i**). Adult traits are shown in (**b**), (**d**), (**f**), and (**h**). Source data are in Supplementary Data 3.

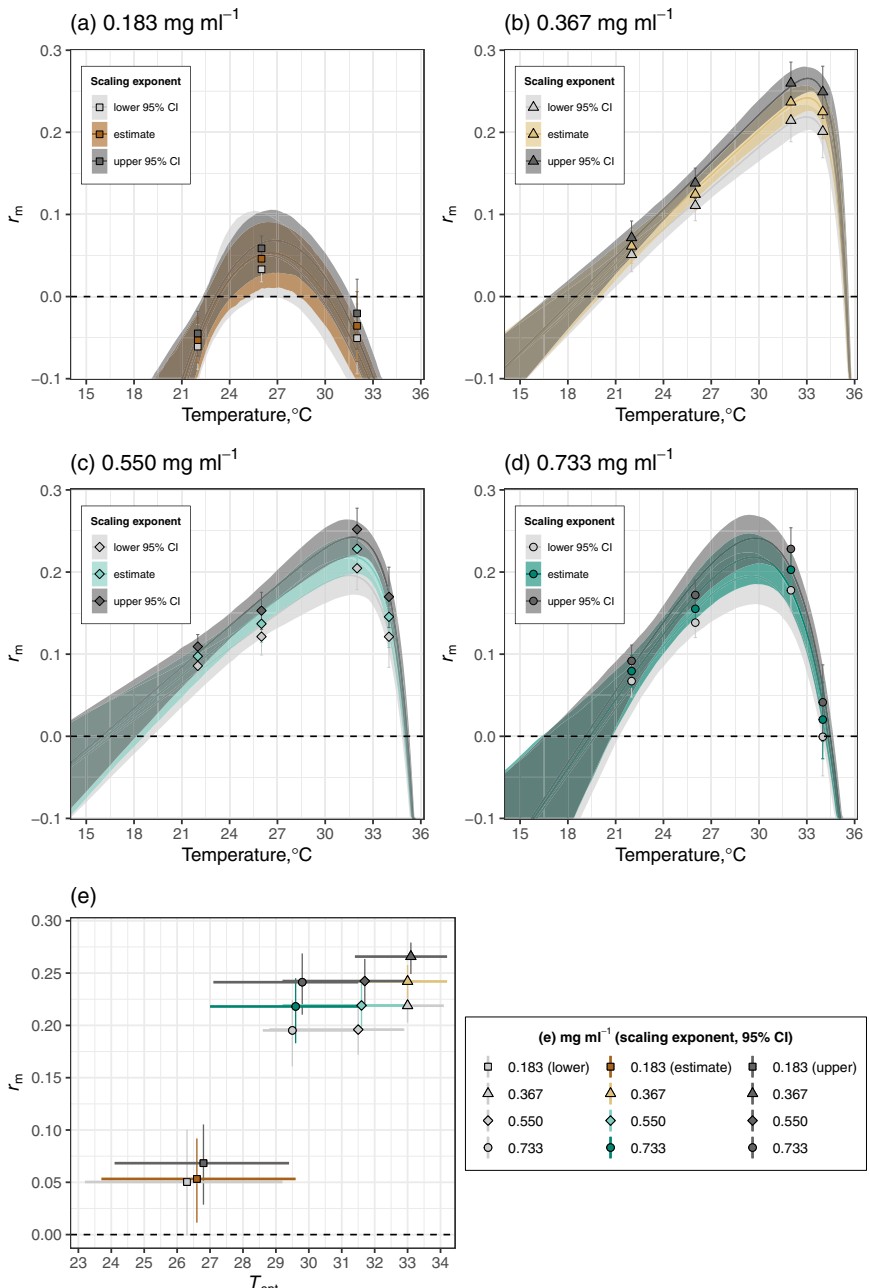

**Fig. 4 The insensitivity of the population fitness ($r_m$) TPCs to uncertainty in our fecundity estimates by resource level. a–d** Comparison with the central estimates (coloured lines and 95% confidence bounds compared with greyscale lines and CBs) shows that, for all resource levels, using the upper and lower 95% exponents (Supplementary Eq. 1, Supplementary Fig. 1b) for the scaling between lifetime fecundity and size does not qualitatively change the predicted $r_m$ TPCs, or the matrix projection $r_m$ estimates (represented by symbols) that were used to fit the $r_m$ TPCs. **e** The insensitivity of predicted $r_m$ $T_{opt}$ to uncertainty in our fecundity estimates by resource level. Using the upper and lower 95% exponents (Supplementary Eq. 1, Supplementary Fig. 1b) for the scaling between lifetime fecundity and size does not qualitatively change predicted maximal $r_m$ (symbols with 95% CIs (vertical, coloured lines)) or $r_m$ $T_{opt}$ (symbols with 95% CIs (horizontal coloured lines)). Source data for the central estimates can be found in Supplementary Data 2; the other source data are in Supplementary Data 4.

However, our results here show that depleting resource environments may indeed still result in a negative effect of competition on $r_m$, even when initial resource levels are high, due to a compounding effect. For example, Huxley et al.[9] found that $r_m$ at high daily per capita resource supply (1 mg larva$^{-1}$ day$^{-1}$) was $0.28 \pm 0.05$ (95% CI) at 32 °C, whereas, it was $0.24 \pm 0.02$ (95% CI) at 32 °C at the optimal resource level (0.367 mg ml$^{-1}$) for $r_m$ in this study. Although this difference in $r_m$ is statistically insignificant, our elasticity analyses here suggest that it derives from

how competition increased development time. At 0.367 mg ml$^{-1}$ in the present study, development time was $6.98 \pm 0.13$ days at 32 °C, whereas, at 1 mg larva$^{-1}$ day$^{-1}$ in Huxley et al.[9] it was $5.81 \pm 0.14$ days at 32 °C.

The trait-level responses at our higher resource levels are congruent with studies that have used the trait responses of optimally fed lab strains to predict how vector fitness and disease transmission will respond to climatic warming. In these studies, mosquitoes are expected to develop at ~0.07 individual$^{-1}$ day$^{-1}$

at 22 °C; increasing to ~0.14 individual$^{-1}$ day$^{-1}$ at 32 °C[28]. In the present study, the development rate (1/development time; Fig. 1b, Table 2) increased by a similar margin when the strength of competition was lessened by high resource availability. In contrast, at low-resource levels, we found juvenile development rate increased from 0.05 day$^{-1}$ at 22 °C to 0.12 day$^{-1}$ at 32 °C (Fig. 1b, Table 2). Although these differences in juvenile development rate may appear small, we show that they can have dramatic effects on the temperature dependence of $r_m$ when combined with the negative impact of intensified competition on juvenile survival (Fig. 1a, Table 2).

Juvenile mortality rate increased significantly with temperature and it was consistently higher at low resource levels (Fig. 1a) than at higher resource levels. This is probably because somatic maintenance costs increase with metabolic rate[44], which cannot be met below a threshold resource level. Intensified competition at low resource levels is also likely to have contributed to preventing some individuals from meeting this increased energy demand. This explains why juvenile mortality rates were highest at 32 and 34 °C at low-resource levels (except at 36 °C where no individuals survived to emergence) where the energy supply-demand deficit was expected to be the largest.

Since larval competition can mediate the temperature dependence of $r_m$, it is also important to determine the temperature dependence of resource availability itself[20]. For example, the natural diet of mosquito larvae comprises of detritus and microbial decomposers[45], which are both sensitive to temperature[46,47]. Therefore, shifts in environmental temperature could alter the concentration of food in the environment, which could affect the growth of detritivore populations. While recent studies have provided useful insights into the relationships between microbes, detritus and mosquito vectors[48–52], future work could focus on the temperature-dependencies of these relationships.

Such a focus could provide important insights into how disease vectors and other arthropods will respond to environmental change. For example, if resource availability increases with climatic warming (e.g., due to increases in decomposition and microbial growth rates), its regulatory effect on population growth and abundance could be relaxed through increased juvenile development and adult recruitment rates. Indeed, increased resource availability with warming could contribute to the expansion of disease vectors and other invasive pest species into regions that were previously prohibitive by broadening $r_m$'s thermal niche width[23,25]. On the other hand, evidence from our high resource level treatments (e.g., a lower $T_{opt}$ at 0.733 than at 0.367 mg ml$^{-1}$) may suggest that warming could have a negative impact on population growth by causing resources to be over-abundant, which could lead to eutrophication and hypoxia in aquatic environments[53].

Alternatively, if climate change reduces resource availability (e.g., by disrupting temperature-dependent consumer-resource relationships), species' spatiotemporal ranges could contract[20,54]. This is because, as we have shown here, intensified competition at low food levels can prevent $r_m$ from being positive at lower temperatures, can lower $r_m$ $T_{opt}$, and can force $r_m$ to become negative at lower temperatures. In this way, the effects of rising temperatures on vulnerable arthropod populations could be especially pernicious, if resource availability is simultaneously reduced[20].

More specifically, accounting for the effects of larval competition at low and depleting food levels on the $r_m$ thermal niche width would significantly alter projections of how climate change will affect vector-borne disease risk through vector populations. For example, the permissible range for DENV transmission by Ae. aegypti was recently projected to be ~18 °C (18–36 °C[28]). In

contrast, when larval resources fall below a certain threshold as we measured here, the effects of competition significantly narrow this thermal niche width to just ~8 °C (23–31 °C, Table 3). This decreased range of temperatures across which positive population growth is expected, is likely to cause dramatic contractions in risk level estimations both spatially and seasonally. Furthermore, it is likely that resource availability is highly variable in the small isolated habitats that mosquitoes typically breed in, which would further decrease the thermal niche width. Thus, our study emphasises the need to develop novel ways of quantifying how resource fluctuations will affect fitness in natural mosquito populations. For example, recently published datasets on the effects of warming temperatures on nitrogen and phosphorus availability[55] could help vector-borne disease models to constrain predictions of resource availability in response to nitrogen deposition, elevated $CO_2$ concentrations and climatic warming.

We used size-scaling to estimate the effect of temperature and resource level on fecundity, because it is anticipated that most of the effect of intensified larval competition at low-resource levels affects adult mosquitoes indirectly by reducing size at emergence and lifespan[56,57]. Despite these assumptions, we show that substantial error in our fecundity estimates would not alter our main conclusions. This is because predicted fitness was relatively insensitive to these traits (Figs. 3 and 4).

Rapid global change is expected to have far-reaching and disruptive ecological impacts[58]. Climate-driven shifts in the spatiotemporal distributions and abundances of organisms are likely to cause widespread harm to ecosystems, biodiversity and society[26,59]. This concern has prompted calls for a more complete understanding of how interactions between environmental factors can affect population-level responses[20,27,60]. Attempts to understand the population-level effects of temperature × resource interactions have focused on prokaryotes or have not considered competition in depleting resource environments. Our study provides rare evidence of how intensified competition below certain resource thresholds can affect the temperature dependence of fitness in a predictable and generalisable way.

## Methods

We investigated the effects of temperature and resource depletion on mosquito life history using a 5 × 4 factorial design comprised of five temperatures (22, 26, 32, 34, and 36 °C) and four resource levels (0.183, 0.367, 0.550 and 0.733 mg ml$^{-1}$). These experimental temperatures span the range of temperatures that this strain of Ae. aegypti (F16-19 originating from Fort Meyer, FL[61]; is likely to experience in the wild between May (the onset of mosquito season) and November[62]. We extended our range to 36 °C to determine the upper critical thermal limit for this strain. Our resource levels are within the range of studies that have investigated the effects of depleting larval resource environments on Ae. aegypti[18]. Our lowest resource level (0.183 mg ml$^{-1}$) was chosen to simulate a level of resource limitation that is expected in natural juvenile habitats[15,16]. Further, our preliminary assays showed that resource levels below 0.183 mg ml$^{-1}$ resulted in complete juvenile mortality.

The experiment was carried out in two randomised blocks. Each block consisted of all five temperatures and two resource levels. On Day 0 of each block, batches of ~800 eggs were deposited into five (one per experimental temperature) plastic tubs containing 300 ml of dechlorinated tap water. We provided each tub with a pinch of powdered fish food (Cichlid Gold®, Hikari, Kyrin Food Industries Ltd., Japan) to prompt overnight hatching. Tubs were randomly assigned to a water bath (Grant Instruments: JAB Academy) set at one of the five experimental temperatures. Water baths were situated in a 20 °C climate-controlled insectary with a 12L:12D photoperiod and 30 min of gradual transition of light levels to simulate sunrise and sunset. On the following day (Day 1), we created the treatments by separating first instar larvae were into cohorts of 50, which were then transferred to clean tubs containing 300 ml of fresh water. Each treatment comprised of three replicate tubs (3 × 50 individuals treatment$^{-1}$). Resource levels were attained by adding 55, 110, 165 and 220 mg of powdered fish food to the tubs, respectively. While natural larval habitats are expected to receive infrequent resource inputs, they are likely to receive some nutritional deposits (e.g., plant material, insects) during the course of a cohort's development period. We attempted to simulate this aspect, and also allow for realistic resource depletion, by allocating food in two pulses. Half of the assigned quantity was provided on Day 1; the remaining half was provided on Day 4. After Day 4, resource levels were not adjusted but water volumes were topped up, if necessary. We also allocated resources in this way because fouling caused

complete juvenile mortality in our preliminary assays when the high resource treatments (165 and 220 mg) received all of their assigned quantities on Day 1 or after Day 4.

**Fitness calculation**. We calculated $r_m$ using a stage-structured matrix projection model (MPM), which describes change in a population over time (Eq. 1[63])

$$\mathbf{N}_{t+1} = \mathbf{M}\mathbf{N}_t \tag{1}$$

where $\mathbf{N}_t$ is a vector of abundances in the stage classes at time $t$ and $\mathbf{M}$ is the projection matrix. The first row of $\mathbf{M}$ is populated with daily fecundity rate (the number of female offspring produced per female at age $i$). The sub-diagonal of $\mathbf{M}$ (Eq. 1) is populated with the survival proportions from age $I$ to age $i + 1$. Multiplying $\mathbf{N}_t$ and $\mathbf{M}$ sequentially across time intervals gives the stage-structured population dynamics. When the stable stage distribution of $\mathbf{N}_t$ is reached, the dominant eigenvalue of the system is the finite population rate of increase ($\lambda$)[63]. The intrinsic rate of population growth is then $r_m = \log(\lambda)$; a population's inherent capacity to reproduce, and therefore a measure of population-level fitness[5,64,65]. Positive and negative $r_m$ values indicate growth and decline, respectively. We used the 'popbio' R package to build and analyse the MPMs[66,67].

**Model parameterisation**
*Immature development time and immature and adult survival proportions*. The survival proportions for the matrix survival elements (the sub-diagonal of $\mathbf{M}$; Eq. 1) were estimated using the 'survival' R package[68]. We defined the juvenile stage duration (i.e., hatching-to-adult) as the mean duration of transitioning into and out of that stage, and a fixed age of adult emergence at the mean age of emergence.

Juvenile development times for each treatment's MPM were predicted using a regression model (detailed in the 'Statistics and Reproducibility section) that was parametrised with individual-level hatching-to-adult times (days). Upon pupation, mosquitoes were transferred to individual falcon tubes containing 5 ml of tap water, which allowed pupa-to-adult development times and the lifespans of individual starved adults to be recorded. In the absence of food, adult lifespan is positively associated with emergent size, so it is a useful indicator of the carry over effects of temperature and competition in the larval habitat[56,69]. Larval development, pupation and mortality (juvenile and adult) were recorded daily.

*Daily fecundity rate*. Fecundity and body size are positively related in many insect taxa, including mosquitoes[70]. For this reason, scaling relationships between fecundity and size are commonly used in predictions of population growth in *Aedes*[71,72]. We provide a detailed description of our method for estimating fecundity in Supplementary Note 1. Briefly, we measured individual dry mass, and estimated lifetime fecundity using previously published datasets on the temperature-dependent scaling between mass and wing length[73], and wing length and fecundity[56,74]. Temperature-specific individual daily fecundity rate is required for the MPMs (Eq. 1), so we divided lifetime fecundity by lifespan and multiplied by 0.5 (assuming a 1:1 offspring sex ratio). Later, we show that this much variation in the scaling of fecundity does not qualitatively change our results.

**Parameter sensitivity**. We used the standard errors of the survival and fecundity element estimates to account for how uncertainty in these traits is propagated through to the $r_m$ estimate[63,75]. For survival, we used the standard errors estimated by the Kaplan–Meier survival function in the 'survival' R package. For fecundity, we calculated the standard errors of the mean daily fecundity rates (Supplementary Table 2) for each treatment using the 'Rmisc' R package[76]. As an additional sensitivity analysis, we recalculated fitness using the upper and lower 95% CIs of the exponents for the scaling of size and lifetime fecundity (Fig. 3).

**Elasticity analysis**. We used elasticities to quantify the relative contributions of individual life history traits to $r_m$. Elasticity, $e_{ij}$, measures the proportional effect on $\lambda$ of an infinitesimal change in an element of $\mathbf{M}$ (Eq. 1) with all other elements held constant (the partial derivative)[77,78]. This partial derivative of $\lambda$, with respect to each element of $\mathbf{M}$, is $s_{ij} = \partial\lambda/\partial a_{ij} = v_i w_j$ with the dot product $\langle \mathbf{w}, \mathbf{v}\rangle = 1$. Here, $\mathbf{w}$ is the dominant right eigenvector (the stage distribution vector of $\mathbf{M}$), $\mathbf{v}$ is the dominant left eigenvector (the reproductive value vector of $\mathbf{M}$), and $a_{ij}$ is the $i \times j$th element of $\mathbf{M}$. Elasticities can then be calculated using the relationship: $e_{ij} = a_{ij}/\lambda \times s_{ij}$. Multiplying an elasticity by $\lambda$ gives the absolute contribution of its corresponding $a_{ij}$ to $\lambda$[77,78]. Absolute contributions for juvenile and adult elements were summed and changed proportionally to quantify the sensitivity of $r_m$ to these traits.

**Statistics and reproducibility**. In the first instance, we used mixed effects models in the 'lme4' R package[79] to test for significant effects of our predictor variables on fitness traits. In the maximal models, temperature × resource level and replicate were fixed effect predictors and block was a random effect. However, the lower AIC scores from versions of the maximal model suggested that variation among replicates within blocks had a non-significant effect on trait responses. Therefore, for normally distributed trait data (adult lifespan and size), we used a full factorial linear regression model (LM) with temperature × resource level and replicate as fixed effects. Model diagnostics provided no evidence to suggest the development

time data were normally distributed, so we used a generalised linear model (GLM) with family = gamma and link = identity.

We tested the effect of resource level on the temperature dependence of daily per capita juvenile mortality rate by fitting an exponential function to the survival data with R package 'flexsurv'[32]. The final mortality model was obtained by dropping terms from the full model (consisting of temperature × resource level + replicate + block as fixed effect predictors). Terms were retained unless their removal worsened model fit (ΔAIC > −2) (Supplementary Table 1). Maximum likelihood methods executed in 'flexsurv' were used to estimate treatment-level juvenile mortality rates and their 95% CIs. Significant effects were interpreted when CIs were nonoverlapping.

**Quantifying the $r_m$ thermal performance curve**. To determine how resource depletion affected the shape of the $r_m$ TPC, we fitted several mathematical models that allow for negative values at both cold and hot extremes, including polynomial models using linear regression, as well as non-linear models with non-linear least squares (NLLS) using the 'rTPC' R package[80]. Overall, the Lactin2[81] and Kamykowski[82] models were equally best-fitting according to the AIC (Supplementary Table 2). From these, we picked the Kamykowski model (Eq. 2) because it was better at describing the estimated $r_m$ at our lowest resource level. This model is defined as

$$r_m(T) = a(1 - e^{-b(T-T_{\min})})(1 - e^{-c(T_{\max}-T)}), \tag{2}$$

where $T$ (°C), and $T_{\max}$ and $T_{\min}$ are the high and low temperatures at which $r_m$ becomes negative, respectively, and $a$, $b$, and $c$, are shape parameters without any biological meaning. Bootstrapping was used to calculate 95% prediction bounds for each $r_m$ TPC[80] and confidence intervals (CIs) around its $T_{opt}$, as well as the thermal niche width ($T_{\max}$ - $T_{\min}$).

**Reporting summary**. Further information on research design is available in the Nature Research Reporting Summary linked to this article.

## Data availability

The datasets generated during the current study are available in Supplementary Data 1–b4.

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

## Acknowledgements

This study was supported by Natural Environment Research Council (NE/L002515/1). We also acknowledge joint Centre funding from the UK Medical Research Council and Department for International Development (MR/R0156600/1).

## Author contributions

K.M., S.P., L.J.C. and P.J.H. contributed to the conception of the study and designed the experiments, L.J.C. provided the mosquitoes; P.J.H. and S.P. performed the modelling; P.J.H. collected the data and analysed it. P.J.H. wrote the first draft of the manuscript, and all authors contributed substantially to revisions.

## Competing interests

The authors declare no competing interests.
