## [Transparent Peer Review File · Communications Biology]

Reviewers' comments:

Reviewer #1 (Remarks to the Author):

This study examines how environmental resource depletion impacts temperature-dependent traits observed in *Aedes aegypti* mosquitoes. This is indeed an important study, as it directly addresses a major gap encountered when applying previous laboratory findings to modeling efforts. The findings of this study could be used to better represent real-world habitat limitations that larval mosquitoes will encounter, thus leading to improved estimates of temperature-dependent outcomes. Incorporating competition effects into mathematical models will enhance their predictive capabilities, and their utility for public health planning and policy formation.

Overall, this was a well written manuscript, and I have no major concerns with general framing and methodology. However, I think that the manuscript would benefit from expanded discussion on certain points, namely contextualizing the findings of this study with other published work.

The authors successfully introduced why current models based on optimal temperature can be biased. How can these results be used to potentially improve those models? Can you comment on how risk estimates may change in light of these findings (i.e. comparison with a previously published model is mentioned, but are these discrepancies enough to say, change risk maps or health advisories resulting from previous models)? Alternatively, is resource depletion too variable in the field to be generalized in a product like a regional model? The effect of competition in larval habitat has the potential to be both highly localized and variable, dependent on resource availability, volume of water in the ovipositional site, mosquito density, predators, etc. Can these effects be reasonably scaled up for a generalized model?

It is mentioned in the discussion that resource depletion/temperature dependent studies have mostly focused on single-celled organisms. To my knowledge, there has been some work done on larval competition dynamics in *Ae. aegypti*, specifically investigating density effects. It seems that food availability and larval density are related concepts, and it would be beneficial to briefly discuss some of that literature, to compare and contrast major findings.

This may be out of scope, but what implications might these findings have for pathogen transmission? For example, would you suspect that carry-over effects from the larval stage would impact adult traits more directly related to transmission (e.g. EIP)?

Reviewer #2 (Remarks to the Author):

Summary: The authors use a factorial experimental design to test the effects of resource levels and temperatures on mosquito traits under conditions with depleting resources. They then use stage-structured models to illustrate how these effects would scale to population growth rates.

I think this is an interesting study, and I find the different thermal performance curves of mosquito population growth rates under different treatment conditions particularly interesting. However, I have a few "major" comments below, the first of which I think is the most important. Without more information, I'm currently not convinced that the experimental conditions of resource pulses on day 1 and day 4 lead to environments that should be thought of as meaningfully experiencing resource depletion.

Major comments:

1) From what I understand, this experiment taking place under 'depleting resource conditions' is what primarily differentiates this study from a recently published study from the same authors (Huxley et al. 2021, Proc B.). Since this is a key factor for why this study is novel, I don't feel that the authors strongly enough 1) defined what they mean by depleting resource conditions, or 2) justified why their experimental conditions represent depleting resource conditions.

The methods state on lines 637-639: "To allow resource depletion, tubs received two pulses of equal quantity. Half of the assigned quantity was provided on Day 1; the remaining half was provided on Day 4." Based on results provided in Table 2, 6/16 treatments had mean development rates less than 8 days in length, and all 12 treatments at 26, 32, and 34 degrees showed mean development rates less than 10.5 days. Is this amount of time (4-6.5 days since the last resource pulse on day 4) enough time for resources to have meaningfully depleted in these tubs, and to make the environments meaningfully different compared to a study that uses smaller, daily or every second day resource inputs? If so, the authors should state this and their evidence for it in the paper.

2) The results are currently a bit difficult to interpret, mostly due to the journal structure of materials and methods at the end of the article. I think the readability would greatly benefit from adding a short paragraph at the beginning of the Results section that very briefly recaps the experiment and statistical methods, even in just a couple sentences. As of now, the first sentence of the Results (lines 173-174) states that "all trait responses varied significantly with temperature and resource level", without ever having introduced what these trait responses are.

3) I had several experimental questions that the methods did not detail. The description of how the experiment was set-up is thorough, but it seems that pertinent information is missing after that. How often were replicates checked to score juvenile development time and survival—daily? Why were resource pulses given on day 1 and day 4, rather than only on day 1 or on day 1 and a different day?

4) Elasticity results (lines 428-436): does the starving of adults in the experiment decrease adult lifespans compared to natural lifespans or compared to experiments that do not starve adults? I'm concerned that if adult starving leads to an underestimate of adult lifespan, this could potentially bias the elasticity results towards underestimating the importance of adult survival on rm.

Minor comments:

Lines 198-199: I think for clarity it should be stated here that this is the larval resource level that is impacting adult lifespan.

Lines 542-544: listing the confidence intervals in parentheses on the two rm estimates (0.28 and 0.24) would be helpful here. Also, is this a useful comparison if the high-resource supply in Huxley et al. 2021 was 1 mg/ml compared to the 0.367 mg/ml here?

Figure 3: I may have missed this, but what is the reason for combining juvenile traits here but separating out adult lifespan and adult fecundity rate? Does this make it an unfair comparison, comparing the effect of both juvenile traits combined versus separate adult traits?

Responses to referees

Reviewers' comments are in **blue**, our responses in **black**.

Reviewer 1's comments

1. Overall, this was a well written manuscript, and I have no major concerns with general framing and methodology. However, I think that the manuscript would benefit from expanded discussion on certain points, namely contextualizing the findings of this study with other published work. The authors successfully introduced why current models based on optimal temperature can be biased. How can these results be used to potentially improve those models? Can you comment on how risk estimates may change in light of these findings (i.e. comparison with a previously published model is mentioned, but are these discrepancies enough to say, change risk maps or health advisories resulting from previous models)? Alternatively, is resource depletion too variable in the field to be generalized in a product like a regional model? The effect of competition in larval habitat has the potential to be both highly localized and variable, dependent on resource availability, volume of water in the ovipositional site, mosquito density, predators, etc. Can these effects be reasonably scaled up for a generalized model?

We have added lines 666-680 to the Discussion to address Reviewer 1's suggestions for discussing how our dataset may be used to potentially improve current and future VBD model predictions and risk estimates. Here, we also (1) discuss how our findings underline the need for novel approaches to measure how resource availability fluctuates in the field, and (2) suggest a potential approach to this problem.

2. It is mentioned in the discussion that resource depletion/temperature dependent studies have mostly focused on single-celled organisms. To my knowledge, there has been some work done on larval competition dynamics in *Ae. aegypti*, specifically investigating density effects. It seems that food availability and larval density are related concepts, and it would be beneficial to briefly discuss some of that literature, to compare and contrast major findings.

We have added to the Discussion (lines 552-559) to address **Reviewer 1's** suggestions for contextualising our study's findings.

3. This may be out of scope, but what implications might these findings have for pathogen transmission? For example, would you suspect that carry-over effects from the larval stage would impact adult traits more directly related to transmission (e.g. EIP)?

We have discussed the potential for the effects of temperature \times resource interactions to affect components of VBD systems to address **Reviewer 1's** suggestion (lines 596-603).

=====

Reviewer 2's Comments

4a) From what I understand, this experiment taking place under 'depleting resource conditions' is what primarily differentiates this study from a recently published study from the same authors (Huxley et al. 2021, Proc B.). Since this is a key factor for why this study is novel, I don't feel that the authors strongly enough 1) defined what they mean by depleting resource conditions, or

We have addressed the first part of this comment (4a) on lines 137-148 by providing our definition of resource depletion.

4b) justified why their experimental conditions represent depleting resource conditions. The methods state on lines 637-639: “To allow resource depletion, tubs received two pulses of equal quantity. Half of the assigned quantity was provided on Day 1; the remaining half was provided on Day 4.” Based on results provided in Table 2, 6/16 treatments had mean development rates less than 8 days in length, and all 12 treatments at 26, 32, and 34 degrees showed mean development rates less than 10.5 days. Is this amount of time (4-6.5 days since the last resource pulse on day 4) enough time for resources to have meaningfully depleted in these tubs, and to make the environments meaningfully different compared to a study that uses smaller, daily or every second day resource inputs? If so, the authors should state this and their evidence for it in the paper.

We describe how the depleting diet treatments were chosen on lines 707-711. We now provide evidence for why these treatments differ from daily inputs on lines 605-615 by comparing the r_m response from our 0.367 mg ml⁻¹ treatment at 32°C to the daily high-resource supply (1 mg larva⁻¹ day⁻¹) at 32°C in our previous study (Huxley *et al.* 2021). Here, we suggest that the difference in r_m at 32°C between the two studies is likely to derive from how competition in depleting resource environments can negatively affect juvenile traits even when initial resources levels are not limited.

5. The results are currently a bit difficult to interpret, mostly due to the journal structure of materials and methods at the end of the article. I think the readability would greatly benefit from adding a short paragraph at the beginning of the Results section that very briefly recaps the experiment and statistical methods, even in just a couple sentences. As of now, the first sentence of the Results (lines 173-174) states that “all trait responses varied significantly with temperature and resource level”, without ever having introduced what these trait responses are.

This is a good point. We have added lines 177-182 to address this comment.

6. I had several experimental questions that the methods did not detail. The description of how the experiment was set-up is thorough, but it seems that pertinent information is missing after that. **[a]** How often were replicates checked to score juvenile development time and survival—daily? **[b]** Why were resource pulses given on day 1 and day 4, rather than only on day 1 or on day 1 and a different day?

We have added lines 765-766 and 724-731 to address parts **[a]** and **[b]** of this comment, respectively.

7) Elasticity results (lines 428-436): does the starving of adults in the experiment decrease adult lifespans compared to natural lifespans or compared to experiments that do not starve adults? I’m concerned that if adult starving leads to an underestimate of adult lifespan, this could potentially bias the elasticity results towards underestimating the importance of adult survival on r_m .

There is very little data available on adult longevity in natural populations, but it is expected that <20% of females live long enough to outlive a pathogen’s extrinsic incubation period (Costero *et al.* 1999; Harrington *et al.* 2001). The EIP for DENV is expected to range between ~14 days at 22°C to ~6 days at 35°C (Mordecai *et al.* 2017), suggesting that starved lifespans from lab-reared populations could be a reasonable approximation of lifespan in the field.

Importantly, our fitness estimates would not have qualitatively changed if we had used non-starved adults rather than starved adults. This is because elasticities quantify the relative change in fitness resulting from a relative change in underlying traits. This means that the relative contributions would be maintained even if non-starved adult lifespans had been used instead. Also, our finding is consistent with fitness studies on disease vectors and other holometabolous insects in showing that fitness is more sensitive to changes in juvenile traits than adult traits. This effect derives from how development time determines the rate of adult recruitment and the onset of reproduction, whereas juvenile survival determines the number of reproducing adults. We incorporate references into the Discussion (lines 578-579) to support our finding.

8) Lines 198-199: I think for clarity it should be stated here that this is the larval resource level that is impacting adult lifespan.

We agree. We have now addressed this on lines 211-215.

9) Lines 542-544: listing the confidence intervals in parentheses on the two r_m estimates (0.28 and 0.24) would be helpful here. Also, is this a useful comparison if the high-resource supply in Huxley et al. 2021 was 1 mg/ml compared to the 0.367 mg/ml here?

We have now included the confidence intervals for both r_m estimates on lines 610-611. Here, we also clarify the distinction between the high-resource supply treatment (1 mg larva⁻¹ day⁻¹) in Huxley *et al.* (2021) and the 0.367 mg/ml treatment in the present study.

10) Figure 3: I may have missed this, but what is the reason for combining juvenile traits here but separating out adult lifespan and adult fecundity rate? Does this make it an unfair comparison, comparing the effect of both juvenile traits combined versus separate adult traits?

Comparing the traits in this way does not bias the results, but we thought this was an important point to clarify and have redone the elasticity analysis to regenerate Figure 3 (line 531). The new figure allows the reader to make a more direct comparison of juvenile and adult trait contributions to r_m . Our key finding of our elasticity analysis—that r_m is more sensitive to changes in juvenile traits than adult traits is unchanged.